# Genomic interrogation of a MAGIC population highlights genetic factors controlling fiber quality traits in cotton

Maojun Wang [1,2,4], Zhengyang Qi[2,4], Gregory N. Thyssen[1], Marina Naoumkina[1], Johnie N. Jenkins [3], Jack C. McCarty [3], Yingjie Xiao[2], Jianying Li[2], Xianlong Zhang[2✉] & David D. Fang [1✉]

Cotton (*Gossypium hirsutum* L.) fiber is the most important resource of natural and renewable fiber for the textile industry. However, the understanding of genetic components and their genome-wide interactions controlling fiber quality remains fragmentary. Here, we sequenced a multiple-parent advanced-generation inter-cross (MAGIC) population, consisting of 550 individuals created by inter-crossing 11 founders, and established a mosaic genome map through tracing the origin of haplotypes that share identity-by-descent (IBD). We performed two complementary GWAS methods—SNP-based GWAS (sGWAS) and IBD-based haplotype GWAS (hGWAS). A total of 25 sQTLs and 14 hQTLs related to cotton fiber quality were identified, of which 26 were novel QTLs. Two major QTLs detected by both GWAS methods were responsible for fiber strength and length. The gene *Ghir_D11G020400* (GhZF14) encoding the MATE efflux family protein was identified as a novel candidate gene for fiber length. Beyond the additive QTLs, we detected prevalent epistatic interactions that contributed to the genetics of fiber quality, pinpointing another layer for trait variance. This study provides new targets for future molecular design breeding of superior fiber quality.

[1] Cotton Fiber Bioscience Research Unit, USDA-ARS, Southern Regional Research Center, New Orleans, LA 70124, USA. [2] National Key Laboratory of Crop Genetic Improvement, Huazhong Agricultural University, 430070 Wuhan, Hubei, China. [3] Genetics & Sustainable Agriculture Research Unit, USDA-ARS, Mississippi State, MS 39762, USA. [4] These authors contributed equally: Maojun Wang, Zhengyang Qi. ✉email: xlzhang@mail.hzau.edu.cn; david.fang@usda.gov

Cotton is the most important natural fiber used in the textile industry, supporting a multibillion-dollar production and processing industry. Cotton belongs to the genus *Gossypium* that includes 45 diploid species ($2n = 2\times = 26$) and 7 tetraploid species ($2n = 4\times = 52$) with different morphology and fiber characteristics[1]. There are four distinct but overlapping stages during cotton fiber development: initiation, elongation, secondary cell-wall biosynthesis, and maturation[2]. Spinnable lint fibers initiate before or on the day of flowering and grow to a final length of about 30 mm. Upland cotton (*G. hirsutum* L.) presently accounts for ~95% of the worldwide cotton production and dominates the world cotton commerce[3]. Consequently, improving the quality of Upland cotton is the main concern in most of the world's cotton breeding programs.

Cotton fiber-quality traits are quantitatively inherited, and influenced by a variety of genes. In order to understand the genetic basis of quantitative traits, many genetic algorithms have been developed[4–6]. A genome-wide association study (GWAS) is a population-based approach that takes advantage of the long history of recombination events in natural populations to identify small haplotype blocks associated with phenotypes of interest across species-scale diversity. Genotyping has been the main limitation of the GWAS method for a long time, but in the past few years, advances in high-throughput sequencing and data processing have facilitated the use of this approach not only in model species, but also in crops[7–14]. However, this approach is limited by the presence of rare alleles and confounding population structure[15].

The algorithms developed for GWAS can also be applied to carefully designed multi-parent populations, where both problems mentioned above can be mitigated to some degree because of balanced allele frequency and controlled population structure[15,16]. Various multi-parent genetic designs have been proposed, including nested association mapping (NAM)[17,18], multi-parent advanced-generation inter-cross (MAGIC)[19,20], random-open-parent association mapping (ROAM)[21,22], and the recently presented complete-diallel design plus unbalanced breeding-like inter-cross (CUBIC)[16]. In plants, a MAGIC population was first created in *Arabidopsis*[19]. Subsequently, more have been created in numerous crop species, including rice[23], wheat[24,25], maize[26], tomato[27], barley[28], and cotton[29,30]. Compared to traditional bi-parental populations, a MAGIC population has a higher potential for more accurate quantitative trait locus (QTL) detection through the segregation of higher genetic diversity, because the progeny lines are mosaics with contributions from all founders. A study in maize showed that a relatively small number of MAGIC lines can achieve a high mapping power[31]. Furthermore, unlike natural populations, the traceability of the pedigree within the MAGIC population makes it possible to perform GWAS based on identity-by-descent (IBD). IBD is a term used in genetics to describe a matching segment of DNA shared by two or more individuals that have been inherited from a common ancestor without recombination. Detection of IBD segments provides a fundamental measure of genetic relatedness and plays an important role in QTL mapping. Studies have shown that IBD-based GWAS is complementary to conventional single-variant-based association mapping and is particularly superior in the identification of QTL with allelic series or small effects[15,16].

For many traits, causal loci uncovered by genetic mapping studies explain only part of the heritable contribution to trait variation. Multiple explanations for this "missing heritability" have been proposed. One of them is the interaction between different loci (epistasis)[32]. Epistatic interactions are the driving factors for the rapid evolution of traits and phenotypic diversification[16,33,34]. However, due to the limitations of genetic and genomic tools and resources, the comprehensive breadth and significance of epistasis

in crop domestication and breeding are not well understood[35]. A wider range of epistatic interactions can be tested in a MAGIC design because a particular haplotype of a founder in one genomic location occurs in combination with the haplotypes of many other founders at different genomic regions[26].

In this study, we performed two complementary GWAS methods (single-variant-based GWAS (sGWAS) and IBD-based haplotype GWAS (hGWAS)) to identify QTLs associated with fiber-quality traits by using a cotton MAGIC population. Beyond the additive QTLs obtained, we also found that epistasis was prevalent, and most epistatic pairs showed moderate effects, indicating that epistatic interactions were as important as additive effects. The full exploration of the genetic architecture of fiber-quality traits will allow genomics-based cotton breeding in the future.

## Results

**Genetic diversity of the MAGIC population**. Eleven cotton cultivars, Acala Ultima (AU), Tamcot Pyramid (TP), Coker 315 (C315), Stoneville 825 (ST825), Fibermax 966 (FM966), M240RNR (M240), Paymaster HS26 (HS26), Deltapine Acala 90 (DP90), Suregrow 747 (SG747), Phytogen PSC355 (PSC355) and Stoneville 474 (ST474) (referred to parent numbers 1-11 in the text, respectively) were crossed in a half-diallel design to produce 550 MAGIC recombinant inbred lines (RILs) ($C_5S_6$)[29,36]. To dissect the genome architecture of this population, we performed resequencing of each parent and RIL. A total of 1,548,294 high-quality SNPs were identified (Fig. 1a and Supplementary Data 1). Among these SNPs, 134,581 (8.69%) were located in gene regions, 79,909 (5.16%) were located in upstream or downstream regions (variant overlaps 1-kb region upstream or downstream of transcription start or end site) and 1,333,804 (86.15%) were located in the intergenic regions. With splicing SNP defined as a variant within 2-bp of a splicing junction, we annotated 16 splicing SNPs in exonic regions and 270 in intronic regions (Supplementary Data 1). In exonic regions, a total of 25,426 (1.64%) non-synonymous SNPs (including stop-gain and stop-loss SNPs) were identified in 12,257 genes (Supplementary Data 2). This rate was much lower than that in rice (4.8%), but was similar to that in maize (1.9%)[37,38]. The SNP distribution along the A and D subgenomes ($A_t$ and $D_t$, in which t indicates tetraploid) was different and the SNP density in $D_t$ was much higher (0.60 SNPs/kb and 0.86 SNPs/kb in $A_t$ and $D_t$, respectively). We found that about 1.56% of SNP genotypes were heterozygous in the 550 RILs, as expected after six generations of self-pollination.

We produced a neighbor-joining (NJ) phylogenetic tree and found that the RILs are distributed at roughly equal distances (Fig. 1b). Using principal component (PC) analysis of the full set of SNPs, we found that the first two principal components only explained 5.64% of the sample variance (Fig. 1c), indicating that the RIL population had been successfully random-mated, and no obvious structure was present. The linkage-disequilibrium (LD) halving distances of the $A_t$ and $D_t$ subgenomes were 1750 and 1200 Kb ($r^2 = 0.33$), respectively (Fig. 1d). The LD decay distance was higher than that in natural populations of Upland cotton[13,39,40]. All the above data indicated that the 550 MAGIC RILs were not structured and had moderate LD, and thus were suitable for GWAS.

**IBD-based mosaic map of MAGIC RILs**. To detect IBD segments, we performed an analysis with a hidden Markov model (HMM) to infer the parental origin of every genomic segment in each RIL. A total of 115,316 IBD segments were identified in 550 MAGIC RILs (Fig. 2a and Supplementary Data 3). The length of the IBD segments per line varied, ranging from 0.1 to 119 Mb, with an average length of 8.12 Mb (Fig. 2b). The mean length of

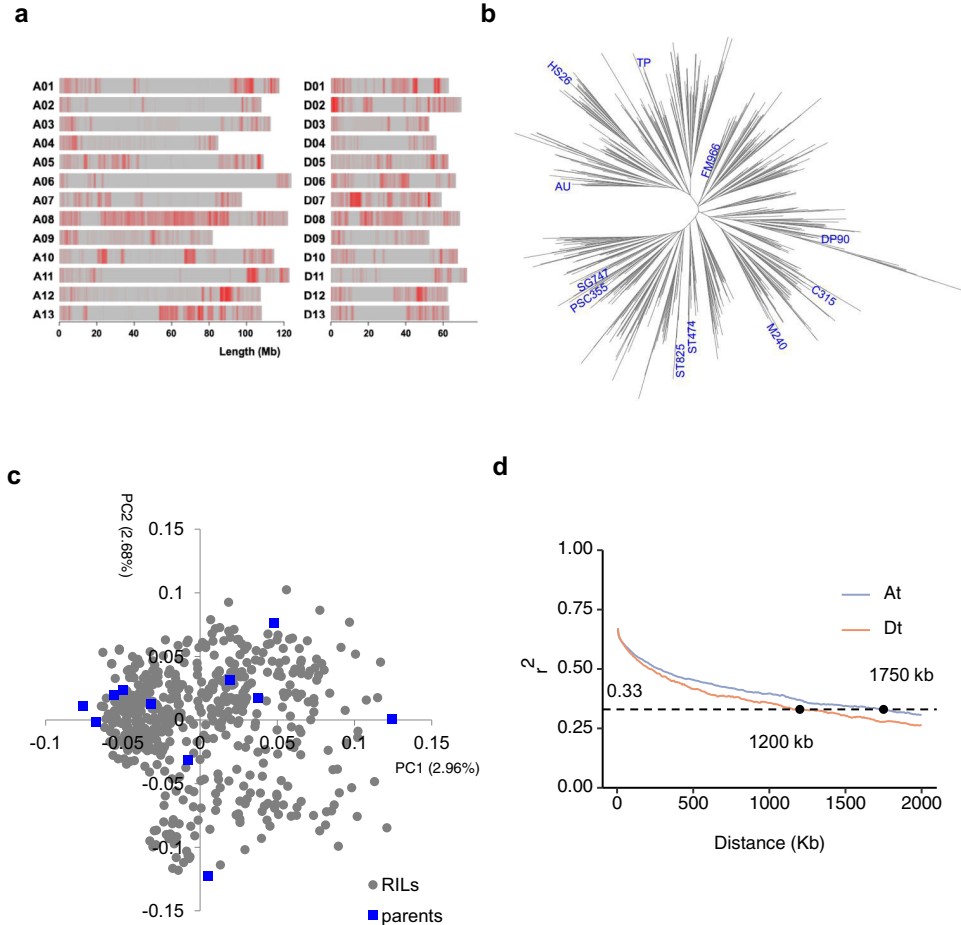

**Fig. 1 Diversity in the cotton MAGIC population. a** Distribution of SNPs in 26 chromosomes. **b** Neighbor-joining tree of the 11 founders and 550 MAGIC RILs, with founder labeled in blue. **c** A principal component analysis of the 11 founders and 550 MAGIC RILs using the full set of SNPs. Blue color represents parents. Gray color represents RILs. **d** Decay of linkage-disequilibrium (LD) in the $A_t$ and $D_t$ subgenomes.

IBD segments was 9.99 Mb and 6.09 Mb in the $A_t$ and $D_t$ subgenomes, respectively. This finding was consistent with $A_t$ being approximately 1.7 times the size of $D_t$[41]. The number of IBD segments per line ranged from 130 to 258, with an average number of 210 (Supplementary Data 3). The mean number of IBD segments was 109 and 101 in the $A_t$ and $D_t$ subgenomes, respectively, indicating the recombination in $A_t$ subgenome was slightly more frequent (Fig. 2c). On average, 23.75% of the DNA fragments were not traced back to any specific parent, presumably reflecting the co-ancestral origins. The eleven parents had different contributions to the genomic composition of the 550 MAGIC RILs. The smallest was FM966 with a contribution rate of 2.3%, and the largest was TP with a contribution rate of 14.3% (Fig. 2d).

Following the IBD analysis, we counted the recombination events in a 2-Mb sliding window across the 550 MAGIC RILs and found that recombination events mostly occurred in the regions far away from the centromeres (Fig. 2a and Supplementary Fig. 1). The average number of recombination events per generation was also counted. On average, there were 38.1 recombination events (1.5 per chromosome) per generation in each RIL, varying from 1 for chromosome (Chr.) D13 to 2.3 for Chr. A05 (Supplementary Data 4).

According to the IBD blocks, a total of 46,204 recombination breakpoints were identified, half of which had only one recombinant event (Supplementary Data 5). However, there were 516 loci in which more than 50 recombination events occurred.

These loci were defined as recombination hot spots. The maximum number of recombination events that occurred in a single 2-Mb sliding window was 360 (Supplementary Data 5). The numbers of recombination hot spots located in $A_t$ and $D_t$ subgenomes were approximately equal (260 and 256 in $A_t$ and $D_t$, respectively).

**Genetic dissection of fiber-quality traits via two approaches.** Two model-based approaches, sGWAS and hGWAS, were used to dissect the genetic basis of fiber-quality traits. These traits comprised fiber elongation (FE), length (FL), strength (FS), uniformity (UI), and micronaire value (MIC) (Fig. 3). In sGWAS, a total of 25 sQTLs were identified, of which 14 were novel QTLs. Detailed information for the five fiber-quality traits across the whole genome is given in Supplementary Data 6. There were 1856 genes in these sQTLs, of which 563 genes had non-synonymous SNPs (Supplementary Data 7). Most detected QTLs had a moderate additive effect, ranging from 0.23 to 0.56 standard deviations for each trait, with an average of 0.36. Among these QTLs, most could explain only a small portion of phenotypic variation, with an average of 7.4%. However, four contributed more than 10% of the trait variance, including two loci for FE on Chr. D04 and D05, one locus for FS on Chr. A07, and one locus for UI on Chr. A07 (Supplementary Data 6). In addition, we found that the QTL on Chr. A07 was associated with both FS and UI, possibly explained by the observed strong phenotypic correlation (Pearson correlation coefficient was 0.74) (Supplementary Fig. 2).

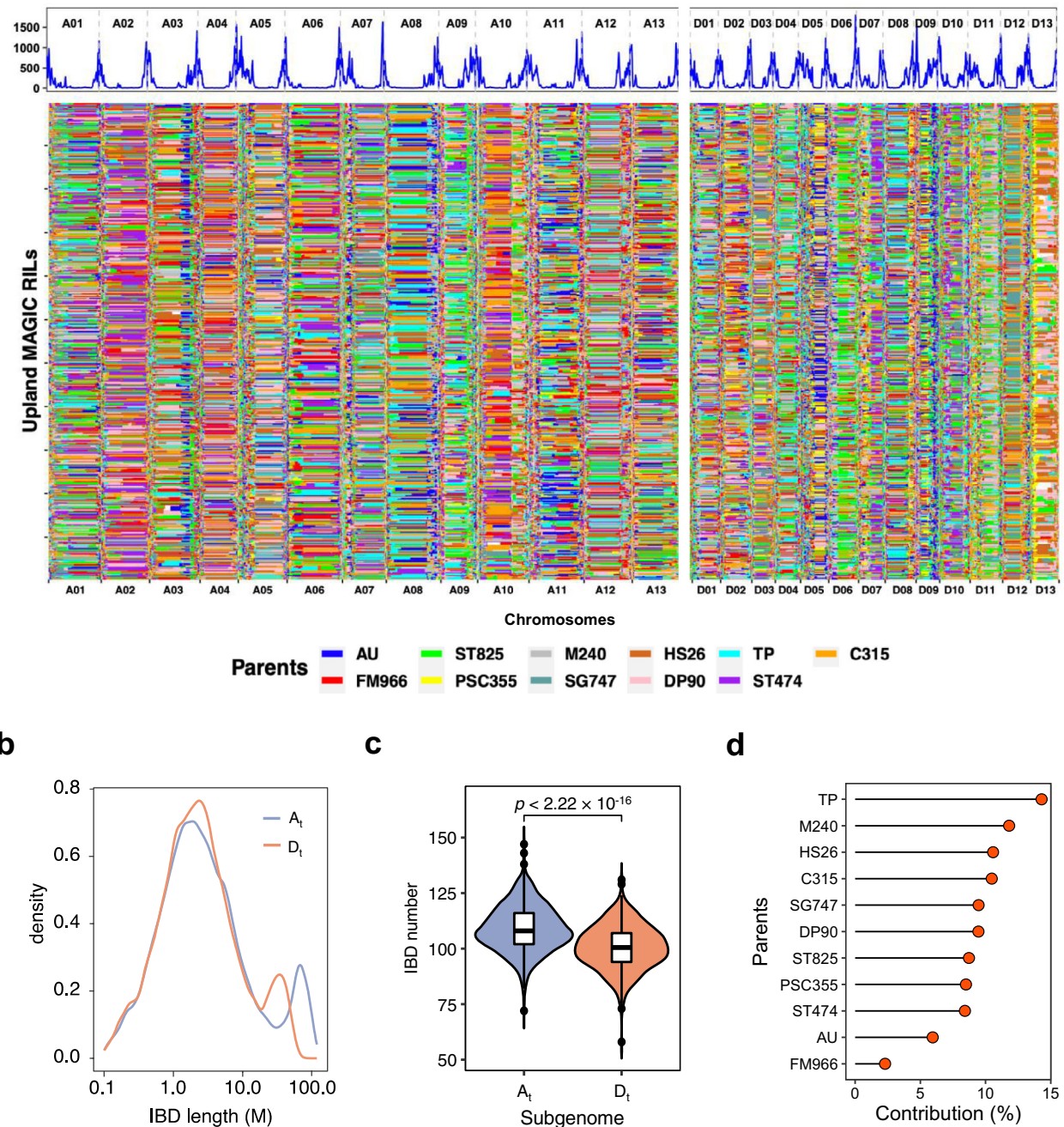

**Fig. 2 IBD-based recombination pattern in the 550 MAGIC RILs across the genome. a** The recombination pattern along the cotton chromosomes (top). Each estimated point indicates the mean value of recombination jointly in a 1-Mb sliding window across the 550 MAGIC RILs. IBD map for 11 founder parents (bottom). Each IBD block is assigned to the parent with the top probability, only if it is larger than 2 folds of expected value by chance (1/11), otherwise, it was considered with an unknown origin (in white). **b** Density plots of the length of IBD blocks in the $A_t$ and $D_t$ subgenomes. **c** Violin plots of IBD number of 550 MAGIC RILs in $A_t$ and $D_t$ subgenomes. *P*-value was calculated by a two-tailed Student's *t*-test. **d** Global contribution of 11 parents in the MAGIC population.

The sQTL jointly explained 23.8% (5–37%) of the phenotypic variation, much lower than the estimated heritability for each trait (Supplementary Data 8).

We reconstructed the founder haplotypes in the MAGIC RILs and performed hGWAS[16] for FE, FL, FS, UI, and MIC. The 95th percentile of 100 permuted likelihood ratio test (LRT) scores was used as the significance threshold. The 37th percentile was used as a suggestive significance threshold[42]. The suggestive threshold

was 10.55, 10.14, 10.35, 10.39, and 10.92 for FE, FL, FS, UI and MIC, respectively. Peaks above the suggestive threshold were identified as hQTL. A total of 14 significant loci were detected in the four cotton fiber-quality traits (FL, FS, UI and MIC), of which 12 were novel QTLs. No significant hQTL was identified for FE (Supplementary Data 9). There were 576 genes in these QTLs, of which 157 had non-synonymous mutations (Supplementary Data 10). These QTLs had small estimated effects with each

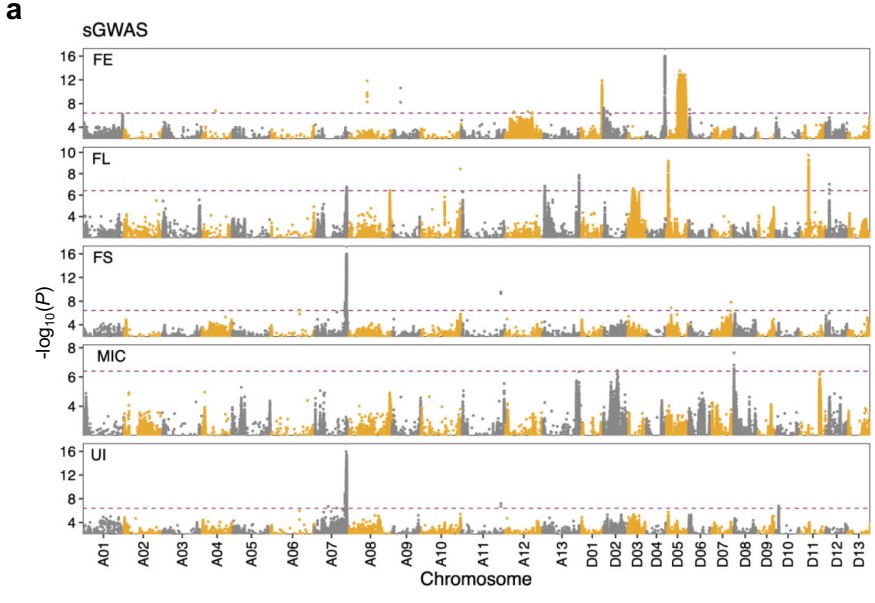

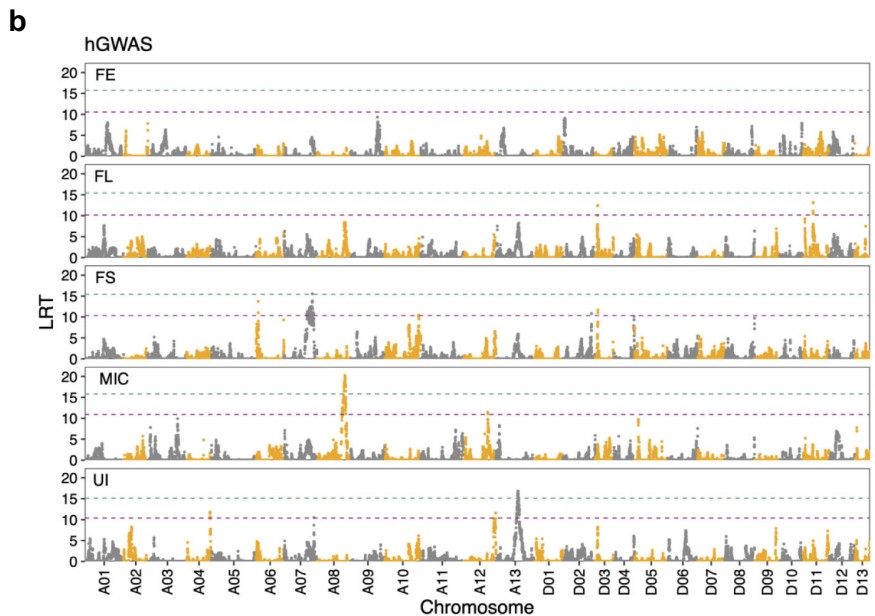

**Fig. 3 Manhattan plots for five fiber-quality traits based on two different GWAS methods. a** SNP-based GWAS (sGWAS). Purple line represents the significant threshold ($-\log_{10}P = 6.43$). **b** IBD-based GWAS (hGWAS). Green line and purple line represent significant and suggestive thresholds. The 95th and 37th percentile of 100 permuted LRT scores were used as the significance threshold (LRT = 15.76, 15.14, 15.43, 15.83, 15.44) and suggestive threshold (LRT = 10.55, 10.39, 10.35, 10.92, 10.14), respectively. The fiber traits include elongation (FE), length (FL), strength (FS), micronaire value (MIC), and uniformity (UI).

explaining a small percentage of the phenotypic variance. The phenotypic variance explained (PVE) by a single hQTL was 3.9% to 10.4%, with an average of 6% (Supplementary Fig. 3a and Supplementary Data 9). There was one hQTL located on Chr. A07 contributed more than 10% of the trait variance for FS. This FS QTL was at the same location identified by sGWAS. Only two sQTL and hQTL were physically co-mapped (Supplementary Fig. 3b), indicating that the two GWAS methods perform in a complementary manner. The two methods identified 10 QTLs that explained above 40% of the total variance for FL and FS, respectively, whereas only 3 QTLs explained 13% of the variance for MIC (Supplementary Data 8). sQTL and hQTL jointly contributed an average of 32.8% of the phenotypic variance (13–41% per trait) (Supplementary Data 8).

**Identification of fiber strength-related genes**. Due to the traceable IBD feature of a MAGIC population, one targeted strategy to identify causal genes is to consider phenotypic differences between individuals with distinct allelic statuses. The MAGIC RILs were first categorized into parental IBD groups according to the peak bin of GWAS. Because some parents may share the same QTL allele, the parental IBD groups with phenotypically distinguishable status were extracted. These IBD states were called functional alleles.

For FS, one large effect QTL was identified on Chr. A07 by sGWAS and hGWAS (Fig. 3). The QTL contributed ~13% of the trait variance, causing a change of 0.97 g/tex. By assessing the IBD status of the peak bin (bin5216, LRT = 15.5), a total of 9 parental IBD groups ($n > 3$) were observed (Fig. 4a, b). We found that the fiber strength of these parental IBD groups was significantly

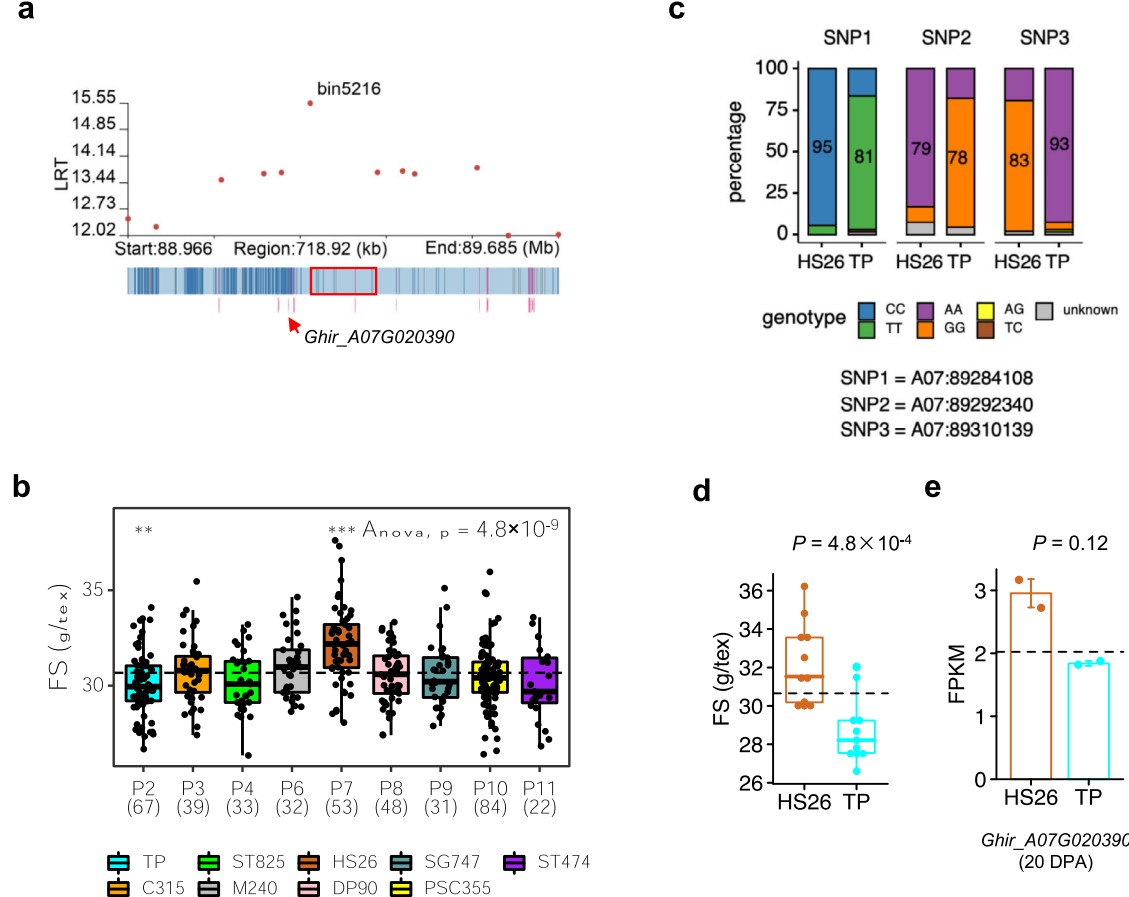

**Fig. 4 Inference of functional allelic types across parent IBD groups for FS on chromosome A07 (88.97-89.69 Mb) and identification of the candidate genes. a** Dot plot of hGWAS result for FS (top) and SNP distribution (bottom) surrounding the peak on chromosome A07. Red box indicates the position of the most significant bin. Arrow indicates the position of candidate gene *Ghir_A07G020390*. **b** Boxplots for FS based on the different parent IBD groups (*P*) (*n* > 3). Parent IBD group was defined by the different status of bin5216. Dashed line indicates the mean value of FS in all RILs. ANOVA (*P* = 4.8 × 10⁻⁹) shows that the mean values of different parent IBD groups are not equal. Differences between RILs of a single parent IBD group and all RILs were analyzed by a two-tailed Student's *t*-test (***$P$ < 0.001; **$P$ < 0.1). The numbers in parentheses are the MAGIC RIL counts in each parent IBD group. **c** Percentage of different genotypes of 3 SNPs located in bin5216 in HS26 and TP. **d** FS of parent HS26 and TP. *P*-value was calculated by Student's *t*-test. Error bars, standard error. **e** Expression of gene *Ghir_A07G020390* in parent HS26 and TP at the cell-wall-thickening stage (20 DPA) of fiber development. *P*-value was calculated by a two-tailed Student's *t*-test. Error bars, standard error.

different (ANOVA, $P = 4.8 \times 10^{-9}$). The fiber strength of parent 7 (P7) descendent RILs was significantly higher than the average of all RILs, and the fiber strength of P2-descendent RILs was significantly lower than the average of all RILs (as determined by two-tailed Student's *t*-test, $P < 0.01$). There were 53 P7-descendent RILs and 68 P2-descendent RILs. By comparing the genotypes of the RILs between P7 and P2, we found three adjacent SNPs in P7 with significantly altered allele frequency compared to P2 (Fig. 4c). The three SNPs were A07:89284108, A07:89292340, and A07:89310139, two of them were not found in sGWAS. This result showed that hGWAS was particularly superior in the identification of QTL with allelic series[16]. The main genotype of A07:89284108 in P7-descendent RILs was CC (95%), but in P2-descendent RILs it changed to TT (81%); for A07:89292340 the change was AA (81%) to GG (78%); for A07:89310139 it was GG (83%) to AA (93%). As some loci were not genotyped because of the relatively low sequencing depth (3× for RILs), the difference between P7- and P2-descendent RILs may be more substantial (Fig. 4c). These three markers can be used to conduct marker-assisted selection breeding.

Particularly, the phenotype of the parents was similar to that of RILs, i.e., the fiber strength of parent 7 was higher than the

average value of all parents, and the fiber strength of parent 2 was lower than the average value of all parents (Fig. 4d). Thus we integrated transcriptomic data of the 11 founders to predict the more likely causal genes within the QTL interval by hypothesizing that the *cis*-effects of the local haplotypic region of each founder on the expression of genes in the nearby region might cause some part of the phenotypic variation. Totally, there were 11 genes in this hQTL region based on the genome annotation information for Upland cotton TM-1. Among these genes, the expression of *Ghir_A07G020390* at 20 days post-anthesis (DPA) showed a positive correlation with the fiber strength (Fig. 4d, e). The previously identified gene *Gh_A07G1769*[13] (corresponding to *Ghir_A07G021030* in our genome) is located in bin5239 (91.43–91.52 Mb). Neither this bin (LRT = 8.9) nor its neighboring bins have reached a significant level (LRT = 10.35), suggesting that this region probably does not contribute much to phenotypic variation.

**Identification of fiber length-related genes.** We identified a QTL in both GWAS analyses on Chr. D11 for FL. This QTL overlapped with those identified in the previous studies[43–45]. According to the

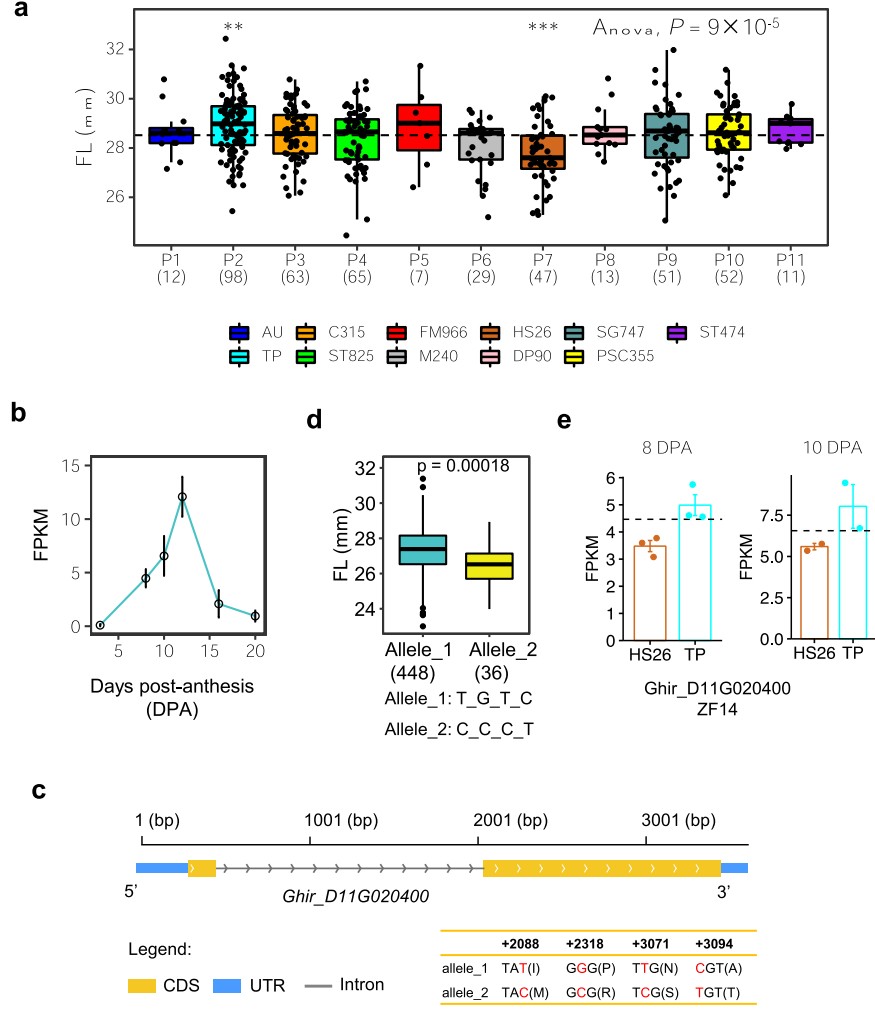

**Fig. 5 Inference of functional allelic types across parent IBD groups for FL on chromosome D11 (24.25-24.72 Mb) and identification of candidate genes. a** Boxplots for FL based on the different parent IBD groups (*P*). Parent IBD group was defined by the different status of bin16702 and bin16703. Dashed line indicates the mean of FL in all RILs. ANOVA ($P = 9 \times 10^{-5}$) shows that the mean values of different parent IBD groups are not equal. Differences between RILs of a single parent IBD group and all RILs were analyzed by a two-tailed Student's *t*-test (***$P < 0.001$; **$P < 0.01$). The numbers in parentheses are the MAGIC RIL counts in each parent IBD group. **b** Expression level of GhZF14 from 3 to 20 days post-anthesis (DPA) during fiber development. Error bars represent s.d. **c** Gene structure of GhZF14. Blue rectangles, yellow rectangles, and black lines indicate UTR, CDS, and intron, respectively. **d** Box plots for fiber length with the two functional alleles shown above (n = 448 vs. 36). Center line, median; box limits, upper and lower quartiles; whiskers, 1.5× the interquartile range. **e** Expression of gene *Ghir_D11G020400* in parent HS26 and TP at 8 DPA and 10 DPA of fiber development.

states of the two most significant bins (bin16702 and bin16703; LRT = 13.1 and 13.0, respectively), the RILs were categorized into 11 parental IBD groups. Each parental IBD group comprised 7-98 RILs. We found that the mean value of fiber length in each parental IBD group was significantly different (ANOVA, $P = 9 \times 10^{-5}$). Parent 7 contributed the short-FL haplotype and parent 2 contributed the long-FL haplotype (two-tailed Student's *t*-test, $P < 0.01$) (Fig. 5a). Within the association region at D11:24.59–24.72 Mb, we identified eleven genes. We integrated the expression-profiling data of parents and functional annotation of the orthologs in *Arabidopsis* to rapidly predict candidate genes associated with FL. In this candidate region, two (*Ghir_D11G020390, Ghir_D11G020410*) of the eleven genes were not expressed, indicating they may be merely silenced in fiber development. Three (*Ghir_D11G020400, Ghir_D11G020420, Ghir_D11G020460*) of the expressed genes had non-synonymous SNPs. Among the three genes, the expression level of *Ghir_D11G020400* (D11:2459616-24603237) showed a dynamic change from 3 to 20 DPA during fiber development, with the highest expression level at 12 DPA (Fig. 5b). This gene was a

homolog of the MATE efflux family protein gene (ZF14) which encodes a plant MATE (multidrug and toxic compound extrusion) transporter that is localized to the Golgi complex and small organelles and is involved in determining the rate of organ initiation in *Arabidopsis*[46]. Four SNPs at the second exon of *Ghir_D11G020400* (named GhZF14) resulted in amino acid changes. D11:24601150 (T/C) at the 2088-bp position in the relevant genome region (195-bp position in the coding sequence (CDS)) resulted in an amino acid change from isoleucine to methionine, D11:24600920 (G/C) at 2318 bp (425-bp in the CDS) resulted in an amino acid change from proline to arginine, D11:24600167 (T/C) at 3071 bp (1178-bp in the CDS) resulted in an amino acid change from asparagine to serine, and D11:24600144 (C/T) at 3094 bp (1201-bp in the CDS) resulted in an amino acid change from alanine to threonine (Fig. 5c). According to these four non-synonymous SNPs, we found two alleles (*n* > 3) in the 550 MAGIC RILs, named Allele_1 (T_G_T_C) and Allele_2 (C_C_C_T). Allele_1 may descend from parent 7 (homozygous in the region) or parent 1, or 2 (heterozygous in the region). The mean FL value of the two haplotypes

showed a significant difference ($P = 1.8 \times 10^{-4}$, two-tailed Student's *t*-test) and Allele_1 increased the value of FL by 0.84 mm compared to Allele_2 (Fig. 5d). Moreover, the fiber length of parent 2 was longer compared to parent 7, with an increased expression level of *Ghir_D11G020400* (Fig. 5e). We noticed that the previously reported KRP6 gene[13,43] was located in the bin16698 adjacent to the significant region identified by hGWAS (bin16699-16703; 24.46–24.72 Mb). However, the LRT value from bin16699 (LRT = 11.1) to bin16698 (LRT = 6.8) dropped rapidly, indicating that there was probably no major gene in bin16698. Considering all evidence together, the gene *Ghir_D11G020400* encoding the MATE efflux family protein was identified as a more likely causal gene for the FL QTL.

**Epistasis serving as a major contributor to trait variance**. Epistasis refers to the interaction between alleles from different loci. Following the identification of QTLs by genome-wide association analysis, we then focused on the detection of effects that might not be identified by using single-locus tests. A total of 581 significant epistatic interactions (epiQTL) were identified for the five fiber-quality traits and most epistatic pairs showed moderate effects, explaining 4% of the phenotypic variations on average (Fig. 6a, b and Supplementary Data 11). This indicated that epistasis was important for fiber-quality trait variance. In detail, there were 195 pairs of epistasis for FE, 52 for FL, 193 for FS, 72 for UI, and 69 for MIC. For FE and FS, nearly 200 significant interacting loci were identified, indicating that epistasis played a more important role in these two traits than other fiber-quality traits. According to the subgenome of interacting loci, the epistasis was classified into three types: AA, AD, and DD. The proportion of DD epistasis was very low in UI (5%) and FE (8%) (Fig. 6a and Supplementary Fig. 4, Supplementary Data 11, 12). The proportion of AD epistasis was relatively higher in FE compared to other traits, probably because an epistatic hotspot existed on Chr. D09 (Fig. 6a). For example, the epistatic effect between two genetic loci for FE—one on Chr. D09 and the other on Chr. A10 is only found when the genotype of D09:35396780 was considered, as there was no difference between the mean FE value of the two genotypes (Fig. 6c). When two loci were considered, the effect of D09:35396780 on the phenotype changed depending on the genotype of A10:58433793 (Fig. 6d).

With the interval of epiQTL defined as the physical position range delimited by the bin, we found that some epiQTLs were linked to significant loci identified by sGWAS for FE, FS, and UI (Supplementary Data 13 and Supplementary Fig. 5). The proportions were 36.31%, 16.67% and 16.92%, respectively (Fig. 6e). For example, A07:91848027 was a significant locus detected by sGWAS for FS. The phenotypic values of the two genotypes (CC and TT) at this locus were significantly different. The mean FS values of CC and TT were 30.36 and 32.04 g/tex respectively. This locus had an epistatic interaction with another locus (A08:48184391; AA/GG), in which the phenotype of plants with the superior allele at A07:91848027 was further enhanced by the genotype of GG at A08:48184391 (i.e., the phenotypic value increased from 30.04 in AATT to 35.09 g/tex in GGTT) (Supplementary Fig. 6). These results indicated some sQTLs may interact with other loci and jointly regulate the development of cotton fiber. Furthermore, we found the recombination frequency in the epiQTL regions was much lower than the random level (Fig. 6f), suggesting that epiQTL regions may be experiencing selection.

## Discussion

One main purpose in plant genetics is to identify the genes (or sequence variants, not all of which are coding genes) responsible for phenotypic variation associated with agronomic traits. A low diversity of the mapping population, small QTL effect, and low frequency of the causal variants are the main factors that influence the comprehensive dissection of the genetic basis of complex traits[47]. The creation of inbred lines derived from multi-parent cross designs has been used to solve these problems. Here we presented an analysis of a MAGIC population in cotton. Conceptually, the MAGIC design requires all parents to be intercrossed in multiple rounds, resulting in a balanced parent composition and numerous recombination events in the offspring, which is critical for improving statistical power and mapping resolution[48]. In our population, the contribution for each parent in genome level ranged between 2.3% for FM966 and 14.3% for TP. The contribution rate was comparable to that reported by Liu et al. (min: 0.5%; max: 13.4%) in maize[16]. In order to boost power for the identification of minor-effect and low-frequency variants, we performed two GWAS methods—SNP-based GWAS and IBD-based GWAS. Few QTLs identified by both statistical approaches were physically co-mapped, indicating that the two GWAS methods worked in a complementary manner. The additive QTLs jointly contributed an average of 32.8% of the phenotypic variance (13–41% per trait). The result was much lower than that in a maize CUBIC population (71%)[16], probably because the overall genetic diversity in cultivated cotton was very low even though these 11 founders were selected to represent the wide spectrum of diversity within the US cotton cultivars[29,49]. Finally, we tested a wide range of epistatic interactions in the MAGIC population and found epistasis was prevalent and significantly contributed to phenotypic variance, indicating that epistasis is important for fiber-quality traits.

The large number of recombination that accumulated during the eleven generations of MAGIC population development ensured a very high resolution to the GWAS mapping approach. In this study, we mapped a major QTL for FL located in D11:24.59–24.72 using IBD-based GWAS. This QTL contributed to 7% of the FL variance, causing a change in fiber length of 0.84 mm. This candidate region containing 11 genes and the likely causal gene *Ghir_D11G020400* (GhZF14) was identified by integrating the expression-profiling data of the 11 parental lines. *Ghir_D11G020400* showed a dynamic change from 3 to 20 DPA during fiber development. Although there has been no functional characterization of *Ghir_D11G020400* in cotton, its homologous gene *AT1G58340* has clear functional information. *AT1G58340* is a member of a subgroup of MATE transporter genes that regulate hypocotyl cell elongation in *Arabidopsis*[46]. Another large effect QTL was identified on Chr. A07 for FS. For this QTL, we used an approach of searching for genes whose transcription patterns matched the founder allele QTL effects, by hypothesizing that the *cis*-effects of the local haplotypic region of each founder on the expression of genes within the same region might cause some part of the phenotypic variation. The expression level of *Ghir_A07G020390* at 20 DPA showed a positive correlation with fiber strength. Thus this gene could be a candidate gene for this QTL.

Although a very large number of genetic variants were identified in our study, only a few were located in genes. It is reasonable to assume that many other genetic variants may have a regulatory role in gene expression. In our previous study, an eQTL hotspot that could regulate the expression of 962 genes was identified by using a natural population of Upland cotton[43]. Interestingly, that eQTL hotspot is very close to the major QTL for FL on Chr. D11 (24.59–24.73 Mb). We also identified several 2-Mb regions where hundreds of recombination breakpoints occurred during the development of the MAGIC population. These regions may have relatively open chromatin structures and possibly be involved in important biological processes[50,51]. Further work will be necessary to investigate the regulatory roles of

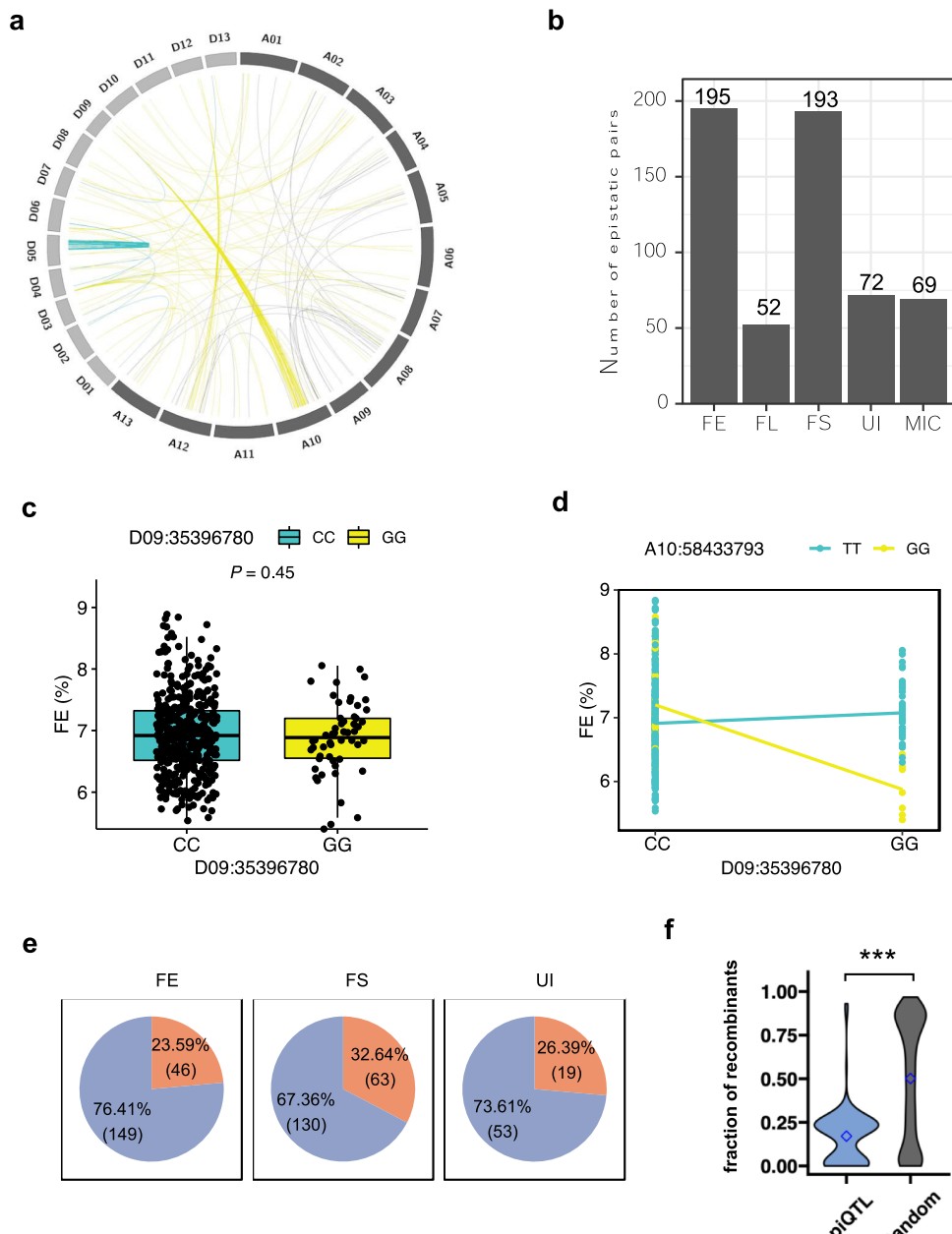

**Fig. 6 Characterization of epistatic contribution to trait variance. a** Significant epistasis for FE. The epistatic analyses for other traits are shown in Supplementary Fig. 4. Gray line, AA epistasis; blue line, DD epistasis; yellow line, AD epistasis. **b** The number of significant epistasis for different traits. **c** Boxplot of FE in two genotypes at SNP D09:35396780 locus. *t*-test shows there is no difference between the two genotypes. **d** Interaction plot for epistasis between two SNPs (D09:35396780 × A10:58433793). This plot displays the relationship between D09:35396780 and A10:58433793 at two values of A10:58433793. **e** Different ratios for identified epiQTL linked with significant SNPs identified by sGWAS. Orange color, epiQTL linked with a significant SNP; blue color, neither of two interacting loci linked with any significant SNP. **f** Fraction of recombinants encompassing interacting pairs of loci between epiQTL and random distal pairs.

these loci. In summary, our work made an effort to fully explore the genetic architecture of cotton quality traits, including additive effect and epistatic interactions, and should facilitate future breeding for fiber quality by pyramiding the elite alleles.

## Methods

**Plant materials**. The MAGIC population consisting of 550 recombinant inbred lines (RILs) was derived from crosses between eleven cotton lines (10 cultivars and 1 breeding line) that represent a diverse group of non-related cotton lines. The details of the population development were described before[29,36,52]. Briefly, the eleven parents were crossed in a half-diallel to establish 55 families. Using 55 $F_1$ crosses as the founding populations, random mating by a bulked pollen approach

was performed for five generations. After five cycles of random mating, self-pollination was followed for six generations using single seed descent. Ten lines were randomly selected from each of these 55 founding populations and a new population including 550 RILs was created.

**Field experiments and phenotyping**. Phenotyping of fiber quality-related traits was performed in 12 natural environments at 4 different locations from 2009 to 2015. The MAGIC RILs and eleven parents were planted in Starkville, MS, in 2009–2011, and 2014–2016; in Florence, SC, in 2014–2016; and in Stoneville, MS, in 2013–2015. Detailed information is given in a previous study[53].

**Estimating breeding value**. The best linear unbiased predictor (BLUP) value for each trait of each RIL was calculated across all replicates, years, and locations using

the mixed linear model in the R package "lme4"[54]. The formula was "model = lmer(phenotype ~ (1|line) + (1|location) + (1|year) + (1|(replicate: location):year) + (1|line: location) + (1|line:year))".

**Genome sequencing**. The 550 MAGIC RILs and their 11 parental lines were grown in a greenhouse in 2013 in New Orleans, Louisiana, USA. Young leaves were collected from 10 plants of each RIL or parent and stored at −80 °C. Genomic DNA was extracted from the frozen leaves using a CTAB method with an additional RNAase A digestion steps as described previously[52,55]. Genomic DNA was sent to Novogene (Chula Vista, CA, USA) for library preparation and the whole genome was sequenced using Illumina HiSeq 2500 with 150 bp end-paired reads. A total of 4.4 TB of sequencing data from 550 RILs (~3× coverage for each) and 550 GB from 11 parents (~20× coverage for each) were obtained[53].

**Mapping and sequence variant calling**. Paired-end resequencing reads were mapped to the Upland cotton TM-1 genome[41] with Burrows–Wheeler-Aligner (BWA) software (the BWA-MEM command) using the default parameters[56]. We used SAMtools to convert the mapping result to the BAM format[57]. The PCR duplicates of sequencing reads for each accession were filtered with the Picard package, and uniquely mapping reads were retained in BAM format. SNP calling was performed using GATK toolkit[58].

**Annotation of genetic variants**. SNP annotation was performed according to the Upland cotton TM-1 genome in the package ANNOVAR[59] (version: 2017-07-16). We performed gene-based annotation to identify non-synonymous SNPs that cause protein-coding changes. SNPs were categorized in exonic regions, intronic regions, and intergenic regions. SNPs in coding exons were further grouped into synonymous SNPs or non-synonymous SNPs; in addition, mutations causing stop-gain or loss were also classified in this group.

**Phylogenetic tree and population structure**. To conduct the phylogenetic analysis, SNPs of all MAGIC RILs were filtered with minor allele frequency (MAF) = 0.05. These SNPs were used to construct a neighbor-joining tree with PHYLIP software and were visualized with the online interactive tree of life (iTOL) tool (https://itol.embl.de/). Principal components analysis (PCA) was performed with this SNP set with the SMARTPCA program embedded in the EIGENSOFT package[60].

**Linkage-disequilibrium (LD) analysis**. The software PLINK[61] was used to calculate the LD coefficient ($r^2$) between pairwise high-quality SNPs; the parameters were set as: '–ld-window-r2 0 –ld-window 99999 –ld-window-kb 1000'. LD decay was calculated on the basis of $r^2$ between two SNPs and averaged in 5-kb windows with a maximum distance of 2 Mb.

**Mosaic map tracing IBD origins**. A hidden Markov Model (HMM) was used to reconstruct the genome of each progeny RIL, since each RIL was made up of IBD segments of the founder genomes. The biallelic SNPs cannot distinguish between all founders, so HMM used the neighboring SNPs to estimate the posterior probability of the offspring line being descended from a given founder. The IBD state is built as the founder with the maximum posterior probability, only if this probability is twice the random probability (1/11); otherwise, it was considered an unknown state[16].

**IBD-based GWAS**. The genome of each MAGIC RIL was divided into 18,003 bins (Supplementary Data 14 and Supplementary Fig. 7) based on all identified recombination breakpoints. In each bin, there was only one IBD state for a given RIL, but 7–11 IBD states were available across all 550 RILs. When calculated, the bins were reformatted as dummy variables. The hGWAS was performed using a mixed linear model, by treating bin and polygenic effects as random effects and the top 10 principal components as fixed effects. The restricted maximum likelihood (REML) was used to test the significance of each bin[16]. The 95th percentile of the permuted LRT scores was chosen as a high significance threshold for each trait. The 37th percentile was used as a suggestive significance threshold for all traits[42]. The suggestive significance threshold ranged from 10.14 to 10.91 for five fiber-quality traits. Significant bins were merged into a locus (or hQTL) with nearby positions (≤1 Mb) or located within ≤5 bins. The interval of each hQTL was defined as the physical position range delimited by the significant bin.

**SNP-based GWAS**. In total, 1,548,294 high-quality SNPs (MAF ≥ 0.05) were used to perform sGWAS. Association analysis was carried out using FAST-LMM (v.2.02) programs[62]. Kinship was calculated based on these SNPs. The significance threshold for the association was set to $6.5 × 10^{-7}$, which was equal to 1/n, where n is the total number of genomic SNPs. To interpret GWAS results, significantly associated SNPs for each trait were first grouped into one locus in which two adjacent SNPs were less than 700 Kb ($r^2 = 0.4$). The consecutive loci were further merged into a single locus if any significant SNPs between adjacent loci were in LD. The significant loci were treated as sQTL, the peak SNP defined the significance of the sQTL, and the extended region of significant SNPs was defined as the sQTL interval.

**Epistasis analysis**. SNPs were first filtered for frequency (MAF ≥ 5%) and linkage-disequilibrium (PLINK 1.9; $r^2 < 0.5$, window size = 50 Kb, step = 5 SNPs). For each trait, the PLINK program (–epistasis) was used to test for SNP × SNP epistasis. As a default setting, pairs of loci with P-value > $1 × 10^{-4}$ was to output. Considering covariate, linear regression with controlling of population structure and additive effect was performed for each putative epistatic pair. Then the P-value was adjusted using Bonferroni correction and those with adjusted P-values < 0.01 were kept as instances of significant epistasis. To estimate the trait variance explained by epistasis, the covariates and additive effects of epistatic pairs were first regressed out, and the residuals for each trait were further regressed against the interacting items. All the calculations were done in R using lm() function.

**Gene expression analysis**. About 50 plants from each of 11 parents were grown during the summer of 2015 in a field in New Orleans, LA. The bolls were collected at five time points, representing early elongation (3 DPA), fast elongation (8, 12 DPA), transition to secondary cell-wall synthesis (16, 20 DPA). Bolls were randomly grouped into three groups to represent 3 biological replicates. The number of bolls per bulked sample varied, with a greater number of bolls required for the earliest time-point. Harvested bolls were placed immediately on ice and transported to the laboratory where they were dissected on ice, frozen in liquid nitrogen, and stored at −80 °C. Total RNA was isolated from detached fibers using the Sigma Spectrum Plant Total RNA Kit (Sigma-Aldrich, St. Louis, MO) as described before[44].

RNA samples from each of the parental lines were sequenced with the paired-end Illumina platform (Platform PE150). The 8 DPA samples were sequenced with three biological replicates, while other DPA samples were sequenced with two replicates. Over 50 million paired raw reads were obtained per sample. Library preparation and sequencing were performed by Novogene Corporation (Chula Vista, CA, USA). The clean reads were mapped to the Upland cotton TM-1 genome with Hisat2 (version 2.1.0)[63]. The expression level of each gene was determined with Stringtie (version 1.3.5)[64].

**Reporting summary**. Further information on research design is available in the Nature Research Reporting Summary linked to this article.

## Data availability

The sequencing data are available in NCBI https://www.ncbi.nlm.nih.gov/, with a Bioproject ID PRJNA789329. The SNP data used in this study are available in https://doi.org/10.6084/m9.figshare.17065469. The authors declare that the other data supporting the findings of this study are available in the Supplementary Information file and Supplementary Data 1–14.

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

## Acknowledgements

This research was funded by the USDA-Agricultural Research Service CRIS projects 6054-21000-018-00D, and Cotton Incorporated project #19-916. We thank Drs. Linghe Zeng in Stoneville, MS and Todd Campbell in Florence, SC for conducting part of field experiments. Mr. Chris Delhom and Mrs. Holly King in New Orleans, LA for measuring the fiber quality attributes using a high volume instrument. The mention of trade names or commercial products in this article is solely for the purpose of providing specific information and does not imply recommendation or endorsement by the USDA, which is an equal opportunity provider and employer.

## Author contributions

D.D.F. conceived and supervised the research, and critically edited the manuscript. M.W. and Z.Q. conducted data analysis and wrote the manuscript. G.N.T., Y.X., and J.L. conducted data analysis. M.N. carried out RNAseq experiment. J.N.J. and J.C.M. developed the MAGIC population. X.Z. supervised data analysis and critically edited the manuscript. All authors critically read, revised, and approved the manuscript.

## Competing interests

The authors declare no competing interests.

**Additional information**

