## [Transparent Peer Review File · Communications Biology]

Reviewers' comments:

Reviewer #1 (Remarks to the Author):

In this study, Wang et al. presented a genomic analysis of a cotton MAGIC population of 550 individuals and identified genetic loci related to fiber quality using two complementary GWAS methods. The merit of this study is to trace the IBD blocks that were used for GWAS analysis. As shown in the manuscript, some novel QTL were found with new candidate genes. The most interesting thing is that the authors performed a genome-wide epistasis study for cotton fiber quality. Although epistasis has long been recognized as an important factor that contributes to phenotypic diversity, its significance is not well understood in cotton, due to the limitations of genetic and genomic tools and resources. This study found that epistasis was prevalent and should be carefully considered during the cotton breeding process. In summary, this study comprehensively analyzed the genetic architecture of fiber quality-related traits and provided a useful resource for future fiber quality improvement. The manuscript is well written, and I only have some minor concerns shown below:

1. Line 95-96: What does "allelic series" mean?
2. Line 310: How do you calculate the phenotypic variations explained by epistatic interactions? The method should be specified.
3. Fig. 6F, Line 339: I want to know how to calculate the recombination frequency and why you said the epiQTL may experience selection.
4. Line 389: Delete "~" in "several ~2-Mb regions", because the sliding window used to count the recombination events is exactly 2 Mb.
5. The statistical tests should be indicated whether one- or two-sided was used (Line 238, 270, 295, 559, 579, 592).
6. Check the grammar and the sentences to make them correct and more clear. Such as Line 55, there should no the "," between family and which; Line 56-57: it is better to say "with morphological variations and differed fiber characteristics."; L59: "which are defined in terms of days post-anthesis (DPA)." Ambiguous description; L64: "Cotton fiber quality traits are quantitative" should be "Cotton fiber quality traits are quantitatively inherited". And so on.....

Reviewer #2 (Remarks to the Author):

The article performed a genetic dissection of fiber quality traits in cotton using a MAGIC population by two GWAS methods, and RNA-seq of 11 parents. The information of the paper is large, and some important and valuable results are obtained. Especially, two major QTLs detected by both methods are responsible for fiber strength and length, and one gene Ghir_D11G020400 (GhZF14) is identified as a novel candidate gene for fiber length. In addition, some epistatic interactions for fiber quality were also detected for trait variance.

Considering the level of this journal, I think the article might be accepted after major revisions.

Certain specific comments are addressed below:

1. The title need to be modified because the genetic in "genetic and epistatic" includes all of genetic component such as additive, dominant, and epistatic effects, etc.
2. In the Abstract, 26 were novel QTLs. The 26 novel QTLs are not reflected or mentioned in the text, please check and add.
3. On line 86, "A study in maize showed that a relatively small number of MAGIC lines can achieve a high mapping power. The mapping power of 500 MAGIC lines was similar to that of 1000 NAM lines 31". Which is the reference in the first sentence? Is it also 31? Please check and supplement.
4. On lines 107-116, some results of this research should not be shown in the Introduction.
5. On line 119, the subtitle "Genetic and phenotypic diversity of the MAGIC population" is improper because there's no phenotype involved in this part.
6. On line 123, it seems that I do not see significant markers 1-11 representing different parents in the text.
7. On line 192-194, "four contributed more than 10% of the trait variance, including two loci for FE on

chromosome D04 and D05, one locus for FS on chromosome A07, and one locus for UI on chromosome A07 (Supplementary Table 6)", only the P value of the QTL for FL on D05 is significant, while the other three are all 0. How to explain? What is "normalized_additive_effect" in Supplementary Table 6?

8. In some parts of Results, it is best not to cite unnecessary literature such as "...of the cell wall" on line 220, and "...in previous studies" on line 231, etc. These can be put into the Discussion.

9. I'm not aware that several replicates of the traits in your study were set up. Please describe experiment design briefly in the text. In addition, if there is any overlap between your study and this study "Whole genome sequencing of a MAGIC population identified genomic loci and candidate genes for major fiber quality traits in upland cotton (*Gossypium hirsutum* L.)"? please answer briefly, no need to add to the text.

10. On line 508, "Cotton bolls were harvested at 3, 8, 12, 16, and 20 days post anthesis (DPA)" please explain why these periods were selected for sampling. Authors can also cite proper references for support.

11. On line 511-512, "3 biological replicates", however, on line 517, "in two biological replicates"?

Responses to Reviewers' Comments:

Reviewer #1 (Remarks to the Author):

In this study, Wang et al. presented a genomic analysis of a cotton MAGIC population of 550 individuals and identified genetic loci related to fiber quality using two complementary GWAS methods. The merit of this study is to trace the IBD blocks that were used for GWAS analysis. As shown in the manuscript, some novel QTL were found with new candidate genes. The most interesting thing is that the authors performed a genome-wide epistasis study for cotton fiber quality. Although epistasis has long been recognized as an important factor that contributes to phenotypic diversity, its significance is not well understood in cotton, due to the limitations of genetic and genomic tools and resources. This study found that epistasis was prevalent and should be carefully considered during the cotton breeding process. In summary, this study comprehensively analyzed the genetic architecture of fiber quality-related traits and provided a useful resource for future fiber quality improvement. The manuscript is well written, and I only have some minor concerns shown below:

Response: Thank you very much for all your suggestions that helped a lot to improve this manuscript.

1. Line 95-96: What does "allelic series" mean?

Response: It means multiple alleles of a QTL. We cited this description from the maize CUBIC population paper (Liu et al. 2020, CUBIC: an atlas of genetic architecture promises directed maize improvement). For example, 4-allelic QTL is expressed as 2

incompletely linked QTN at a given QTL in that paper. This is the phenomenon in which different variations at the same locus lead to the same or very similar phenotypes.

2. Line 310: How do you calculate the phenotypic variations explained by epistatic interactions? The method should be specified.

Response: The calculation was very similar to the calculation of phenotypic variations explained by additive effect. The only difference is that we thought about the epistasis effect when we partitioned the observed variance of phenotypes into components attributable to different sources of variation. In details, we first regressed out the covariates and the additive effect of significant epistatic pairs and then we used the residuals to calculate the percentage of the variance explained by epistatic interaction. All the calculations were done in R using `lm()` function. The method was mentioned in the manuscript, but it may not be clear. We have changed the description of the method and added some details in the Methods section Page 20 of the revised manuscript).

3. Fig. 6F, Line 339: I want to know how to calculate the recombination frequency and why you said the epiQTL may experience selection.

Response: Before calculating the recombination frequency, we used the hidden Markov Model to infer every genomic segment of each offspring descended from any given parent. In this case, we can know whether an epistatic pair underwent recombination in each offspring or not. This was described in the Methods “Mosaic map tracing IBD origins”. The recombination frequency is the ratio of recombined offsprings to all offsprings. The simulated epistatic pairs match the length and number of the real epistatic pairs. We found that epistatic pairs are more likely to maintain the original combination in the offspring, indicating that such a combination is functional and may have undergone selection.

4. Line 389: Delete "~" in "several ~2-Mb regions", because the sliding window used to count the recombination events is exactly 2 Mb.

Response: We have corrected this in the revised manuscript.

5. The statistical tests should be indicated whether one- or two-sided was used (Line 238, 270, 295, 559, 579, 592).

Response: Thank you for pointing this. The issue is corrected in the revised manuscript.

6. Check the grammar and the sentences to make them correct and more clear. Such as Line 55, there should not be the “,” between family and which; Line 56-57: it is better to say “with morphological variations and differed fiber characteristics.”; L59: “which are defined in terms of days post-anthesis (DPA).” Ambiguous description; L64: “Cotton fiber quality traits are quantitative” should be “Cotton fiber quality traits are quantitatively inherited”. And so on.....

Response: Thank you for your suggestions. We have re-written the sentence in the revised manuscript as the following: Line 55, we deleted “in the *Malvaceae* family” to avoid any confusions. Line 56-57, we rewrote the sentence as “with different morphology and fiber characteristics”. Line 59, we have deleted the clause “which are defined in terms of days post-anthesis (DPA)” to make the meaning clearly. Line 64, we have changed the description to “Cotton fiber quality traits are quantitatively inherited”.

Reviewer #2 (Remarks to the Author):

The article performed a genetic dissection of fiber quality traits in cotton using a MAGIC population by two GWAS methods, and RNA-seq of 11 parents. The

information of the paper is large, and some important and valuable results are obtained. Especially, two major QTLs detected by both methods are responsible for fiber strength and length, and one gene Ghir_D11G020400 (GhZF14) is identified as a novel candidate gene for fiber length. In addition, some epistatic interactions for fiber quality were also detected for trait variance.

Considering the level of this journal, I think the article might be accepted after major revisions.

Response: Thank you very much for all your suggestions that are very helpful to improve this manuscript.

Certain specific comments are addressed below:

1. The title need to be modified because the genetic in “genetic and epistatic” includes all of genetic component such as additive, dominant, and epistatic effects, etc.

Response: We are grateful for the suggestion. The title has been modified as "Genomic dissection of a MAGIC population highlights genetic factors controlling fiber quality traits in cotton".

2. In the Abstract, 26 were novel QTLs. The 26 novel QTLs are not reflected or mentioned in the text, please check and add.

Response: Thank you for the suggestion. We have added the necessary information in the Results section (Line 198, 218). This information was provided in the Supplementary Table 6 and 9.

3. On line 86, “A study in maize showed that a relatively small number of MAGIC lines can achieve a high mapping power. The mapping power of 500 MAGIC lines was similar to that of 1000 NAM lines 31”. Which is the reference in the first sentence? Is it also 31? Please check and supplement.

Response: The reference of the first sentence is also 31. To be more clearly, we delete the last sentence in the revised manuscript.

4. On lines 107-116, some results of this research should not be shown in the Introduction.

Response: Thank you for your suggestion. We have re-written the last paragraph of the Introduction section.

5. On line 119, the subtitle “Genetic and phenotypic diversity of the MAGIC population” is improper because there's no phenotype involved in this part.

Response: Thank you for your suggestion. We have deleted the word *phenotypic* in the subtitle.

6. On line 123, it seems that I do not see significant markers 1-11 representing different parents in the text.

Response: We used these numbers to represent the 11 parents for brevity, for example, P1 refers to Acala Ultima, P7 for HS26. In the *Identification of fiber-strength-related genes* section, such as “The fiber strength of parent 7 (P7) descendant RILs was significantly higher than the average of all RILs, and the fiber strength of P2-descendent RILs was significantly lower than the average of all RILs (as determined using two-tailed Student’s t test, $P < 0.01$). There were 53 RILs in P7 and 68 RILs in P2.”. These numbers were also used in the section *Identification of fiber-length-related genes* and Figure 5.

7. On line 192-194, “four contributed more than 10% of the trait variance, including two loci for FE on chromosome D04 and D05, one locus for FS on chromosome A07,

and one locus for UI on chromosome A07 (Supplementary Table 6)”, only the P value of the QTL for FE on D05 is significant, while the other three are all 0. How to explain? What is “normalized_additive_effect” in Supplementary Table 6?

Response: We used the FaST-LMM to perform the SNP-based GWAS. The original p value of these three peak SNPs is "0.0000000000000000E+00" (a format of double precision type numeric data). "0.0000000000000000E+00" is very very close to 0, so they are all significant.

The normalized additive effect is defined as the additive effect divided by the standard deviation of each specific trait. The value is independent of the traits' units of measurement, which allows direct comparisons. We have added the explanation in Supplementary Table 6.

8. In some parts of Results, it is best not to cite unnecessary literature such as “...of the cell wall” on line 220, and “...in previous studies” on line 231, etc. These can be put into the Discussion.

Response: Thank you for your suggestion. We have deleted the unnecessary literature in the revised manuscript.

9. I'm not aware that several replicates of the traits in your study were set up. Please describe experiment design briefly in the text. In addition, if there is any overlap between your study and this study “Whole genome sequencing of a MAGIC population identified genomic loci and candidate genes for major fiber quality traits in upland cotton (*Gossypium hirsutum* L.)”? please answer briefly, no need to add to the text.

Response:

Thank you for the suggestion. We have added a brief description in the *Field experiments and phenotyping* section. Also, the order of this section has been adjusted to make the experimental design clearer for readers.

We used the same phenotype and genotype data as the paper you mentioned. But in this paper, we mainly focused on the IBD identification as well as the IBD-based GWAS. And another aim was to study the contribution of epistasis to the cotton fiber quality traits.

10. On line 508, “Cotton bolls were harvested at 3, 8, 12, 16, and 20 days post anthesis (DPA)” please explain why these periods were selected for sampling. Authors can also cite proper references for support.

Response: Thank you for your suggestion. There are four overlapping stages during fiber development: initiation, elongation, secondary cell wall (SCW) biosynthesis, and maturation, which are defined on the basis of the number of days post anthesis (DPA). This sentence was changed to “The bolls were collected at five time points, representing early elongation (3 DPA), fast elongation (8, 12 DPA), transition to secondary cell wall synthesis (16, 20 DPA).” We collected bolls that can approximately cover the whole fiber elongation stage (3, 8, 12, 16 and 20 DPA).

11. On line 511-512, “3 biological replicates”, however, on line 517, “in two biological replicates”?

Response: We apologize for the mistake. For the 8 DPA, there were 3 biological replicates for the 11 parents. And for other four developmental stages, there were 2 biological replicates. We have made corrections in the revised manuscript.

REVIEWERS' COMMENTS:

Reviewer #2 (Remarks to the Author):

I have reviewed the revised manuscript. I think the authors have revised it according to the revised suggestions. This article can be accepted and published.